# The Goat as a Risk Factor for Parasitic Infections in Ovine Flocks

**DOI:** 10.3390/ani11072077

**Published:** 2021-07-12

**Authors:** David García-Dios, Rosario Panadero, Pablo Díaz, Miguel Viña, Susana Remesar, Alberto Prieto, Gonzalo López-Lorenzo, Néstor Martínez-Calabuig, Pablo Díez-Baños, Patrocinio Morrondo, Ceferino M. López

**Affiliations:** INVESAGA Group, Departamento de Patoloxía Animal, Universidade de Santiago de Compostela, Avda. Carballo Calero s/n, 27002 Lugo, Spain; david.garcia.dios@rai.usc.es (D.G.-D.); rosario.panadero@usc.es (R.P.); pablo.diaz@usc.es (P.D.); miguel.acivo@gmail.com (M.V.); susana.remesar@usc.es (S.R.); alberto.prieto@usc.es (A.P.); gonzalo.lopez.lorenzo@gmail.com (G.L.-L.); nestor.martinez@usc.es (N.M.-C.); pablo.diez@usc.es (P.D.-B.); patrocinio.morrondo@usc.es (P.M.)

**Keywords:** sheep, goat, mixed management, risk factor, parasitic infection

## Abstract

**Simple Summary:**

Small ruminants in northwestern Spain are frequently managed in mixed flocks. Sheep-goat joint management has not been considered a risk factor for parasite infection, so the main objective of this study was to establish if mixed management with goats supposes a risk factor for parasitic infections in ovine flocks. Goat contact was a risk factor for sheep to be infected by protostrongylids, *Dictyocaulus filaria*, gastrointestinal nematodes and *Eimeria* spp. In relation to host-specific parasites, goats cannot be considered as a source for sheep, but competition for food and spaces between both small ungulates can suppose a reduced grazing area to sheep, provoking high environmental contamination and stress that facilitate their infection.

**Abstract:**

Small ruminants in northwestern Spain are frequently managed in mixed flocks. Sheep–goat joint management has not been considered a risk factor for parasite infection, so the main objective of this study was to establish if mixed management with goats supposes a risk factor for parasitic infections in ovine flocks. Two thousand and ninety-three sheep were sampled from 74 commercial meat ovine flocks for diagnostic of the main parasites. Goat contact was a risk factor for sheep to be infected by protostrongylids, *Dictyocaulus filaria*, gastrointestinal nematodes and *Eimeria* spp. In contrast, *Moniezia*, *Fasciola hepatica*, *Dicrocoelium dendriticum* and Paramphistomidae prevalences were not influenced. Sheep–goat mixed management can be considered as a risk factor, since goats would act as a source of pasture contamination for interspecific parasites (protostrongylids, *Dictyocaulus filaria* and gastrointestinal nematodes). In relation to host-specific parasites, such as *Eimeria* spp., goats cannot be considered as a source for sheep, but competition for food and spaces between both small ungulates can suppose a reduced grazing area to sheep, provoking high environmental contamination and stress that facilitate their infection. Future epidemiological studies for parasitic infections in small ruminants should consider sheep–goat mixed management as a possible risk factor to be included in multivariate analyses.

## 1. Introduction

Northwestern Spain is an important cattle breeding area. Although the dairy industry constitutes the main economic activity of the agricultural sector, other species, such as sheep and goats, are achieving great impact in the sustainability of rural communities (economic maintenance and profitability of local farms). Small ruminants make it possible to take advantage of underutilized soil for grazing cattle, fixing population in rural areas and controlling the biomass and fires in forests, thus avoiding environmental degradation. In Galicia, there are 21,709 small ruminant flocks, with a mean of 10.6 animals and goat livestock has been always intimately related with sheep; sheep–goat mixed flocks represent 12.3% of the small ruminant farms, but include 24.6% of the sheep population in Galicia (45,627 out of 185,236) and almost half of the goats (21,873 out of 44,049) [1].

Sheep–goat joint management has not been considered a risk factor for parasite infection so far, although both species share many parasites. A previous study of ovine chronic respiratory diseases revealed that contact with goats, even at low levels, had constituted a risk factor for protostrongylid infections in sheep [2]; however, there is no information about other parasitic infections. The consideration of mixed management with goats as a risk factor for sheep infection has also been found in viral infections; high contact between sheep and goats has been shown as a risk factor for Visna-Maedi virus (VMV), even in flocks with a low number of goats [3].

The main objective of this study was to establish if sheep–goat mixed management supposes a risk factor for the main digestive and respiratory ovine parasitic infections.

## 2. Materials and Methods

### 2.1. Ethics Approval Statement

All faecal samples used in this study were collected with the permission of the farm owners. All experimental procedures fully complied with European and Spanish ethics regulations on the protection of animals used for scientific purposes (European Directive 2010/63/EU and Spanish Royal Decree 53/2013) and approved by the ethical committee of the University of Santiago de Compostela.

### 2.2. Animals and Flocks Surveyed

The study was carried out in Galicia, northwest of Spain (41°49′–43°47′ N, 6°42′–9°18′ W), covering an area of 29,574 km^2^, with mild annual temperatures (13–14 °C) and moderate annual rainfall (1300–1500 mm).

In Galicia, small ruminants are mainly maintained in a semi-extensive husbandry system, with pastures near the barn where they are kept at night, and all farms used anthelmintic treatment to control gastrointestinal nematodes. A total of 2093 sheep fecal samples were recovered from 74 commercial meat ovine flocks (Figure 1); 58 were sheep-pure flocks (1625 sheep) and 16 were sheep–goat mixed flocks (468 sheep). In the latter, 25% of goats (103 goats) were also sampled.

All the animals were mixed breed and older than 6 months; sampled flocks were distributed over the entire region. Sampled animals had not received an anthelmintic treatment since the previous campaign, that is, animals were sampled at least two months after the last treatment.

### 2.3. Examination of Fecal Samples

Fecal samples were collected directly from the rectum with plastic gloves and kept at 4 °C until being analyzed by different coprological techniques (flotation, sedimentation and migration) on the same day. Briefly, for the Baermann–Wetzel migration technique [4,5] to detect protostrongylid (mainly *Muellerius capillaris*, but for four sheep with *Neostrongylus linearis*) and *Dictyocaulus filaria* larvae, 10 g of feces were left overnight with non-woven filters (Filter-Lab, Filtros Anoia, S.A., Barcelona, Spain) in glass funnels with a 12 mL centrifuge tube at the end of a silicone tube. Tubes were centrifugated 10 min at 350× *g* and larval population was retained in the last ml to be analyzed in Favati chambers. These methacrylate counting chamber present a cell of 24 × 24 × 3 mm, enabling the microscopic observation of the volume resulting from centrifugation; the cell contains a permanent grid that allows the user to count more accurately. For gastrointestinal nematodes (GIN–strongyle type eggs), cestode eggs and *Eimeria* oocysts, a flotation technique (modified McMaster method) [6] was used. Firstly, 3 g of feces were weighed and added to 42 mL of water and glass balls (6 mm diameter) to mix the contents thoroughly. This material was then filtered through a 150 µm sieve (CISA Sieving Technologies, Barcelona) to two glass tubes that were filled with the fecal suspension. The tubes were centrifugated at 500× *g* for 5 min and supernatant was discarded. Tubes were then filled with saturated NaCl flotation solution (SG 1.19 at 20 °C) and sediment was gently mixed. Parasitic forms were counted in McMaster chamber; the number of eggs per g of feces is obtained by multiplying the total number of eggs in the two squares by 50 (3 g of feces yielded 45 mL of suspension and 0.3 mL was examined) [6]. Trematode eggs (*Fasciola hepatica*, Paramphistomidae and *Dicrocoelium dendriticum*) were detected by using the sedimentation technique. Feces were weighed (3 g) into a plastic container with 40–50 mL of tap water and glass balls. The solution was mixed thoroughly and filtered through a 150 µm sieve to a plastic cup cleaning the sieve until 1 l is deposited in the conical cup. The mixture sedimented for 20 min and the supernatant was very carefully removed until 200 mL remained. This 200 mL were resuspended to 500 mL and allowed to sediment for a further 20 min. The supernatant was removed again until 50 mL and resuspended to 100 mL and allowed to sediment for a further 20 min, and the supernatant was eliminated until 50 mL remained. Finally the last 50 mL was mixed and observed an aliquot was observed in the bottom of a McMaster chamber.

Morphological diagnostic techniques were performed in strict order: first, feces were weighed to migration, then to flotation and finally for sedimentation, so there were 2093 migrations, and 1914 flotation and 1697 sedimentation tests, according to the feces available.

### 2.4. Statistical Analysis

The associations between sheep–goat contact and the prevalence of parasitic infections were studied individually with a Chi-squared test, with the cc() function, included in the epiDisplay package [7,8] in R statistical software (R v.3.5.3) [9]. The cc() function produces an odds ratio, calculated with the exact method, 95% confidence interval, Chi-squared and Fisher’s exact tests. Then, Partial Least Square Path Modeling [10] was used to build a multivariate approach to the analysis. PLS-PM can handle single and multi-item constructs, making it possible to explore the associations between the parasitic infection variables and the sheep–goat contact (using the plspm() function in the plspm package) [10].

For the examination of the intensity of infection, a non-parametric Wilcoxon rank test was applied to positive sheep depending on whether they were in contact with goats or not, considering that fecal counts in parasitic forms are not normal (shapiro.test() function, R v.3.5.3) [9].

## 3. Results

### 3.1. Prevalence and Intensity of Infection in Sheep and Goats

Table 1 show the overall prevalence and intensity of infection for sheep and goats included in this study. Coccidia were the most prevalent parasites for both species, followed by GIN in sheep and protostrongylids in goats. Goats presented higher prevalence for protostrongylids, *D. filaria*, GIN and *Eimeria* spp. infection than sheep, and also higher outputs but for *D. filaria*; this was especially relevant for protostrongylids with percentages six times higher in goats than in sheep. No goat was positive to trematode or *Moniezia* infection; these parasites were only present in sheep in very low prevalence.

### 3.2. Effect of Sheep–Goat Contact on Parasitic Prevalences

Individual sheep prevalences for the different parasitic infections in pure and mixed flocks are shown in Table 2.

In general, prevalences for the different parasites were higher in mixed than in pure sheep flocks. Chi-squared test indicated that goat contact was a risk factor for sheep to be infected by protostrongylid nematodes, *D. filaria*, GIN and *Eimeria* spp. (*p* < 0.001) in all cases. For those infections, prevalence was from 1.61 to 1.99 times higher in sheep from mixed flocks than in ovine in pure flocks. On the other hand, prevalences for *Moniezia*, *F. hepatica*, *D. dendriticum* and Paramphistomidae were not influenced by goat contact. Multivariate analysis showed the same associations indicated by univariate Chi-squared test; correlations obtained with the plspf() function between infection variables and the sheep–goat contact are displayed in Figure 2. In order of relationship, the most affected infection was coccidia, followed by GIN, protostrongylid and *D. filaria* infection, among the significantly related ones. Although significant, R values were low, infections were also correlated among them, mainly with GIN and *D. filaria* (data not shown), and the effect of mixed management is only one of the risk variables for the sheep infections.

### 3.3. Intensity of Infection and Goat Contact Effect

Sheep parasitic outputs are shown in Table 3. Wilcoxon rank test indicated differences between the two groups only in protostrongylid infection (15.9 larvae per gram of feces—lpg—in the goat contact group vs. 9.8 lpg in pure sheep flocks) and Paramphistomidae (168.3 eggs per gram—epg—in the goat contact group vs. 62.8 epg in pure sheep flocks). In the case of Paramphistomidae, the results should be interpreted with caution as only three animals that were in contact with goats, and nine that were in pure flocks, were positive.

## 4. Discussion

In this study, the most prevalent parasites in both sheep and goats, under a semi-extensive management system, were coccidia, GIN and lungworms. For all those parasites, the prevalence and intensity of infection were higher in goats than in sheep; those differences were especially relevant for protostrongylids. Other previous studies have revealed higher prevalences [11,12] and larval outputs [11,13] in goats than sheep. The same situation has been observed for GIN infection, both in experimental [13,14,15] and natural infections [16]. In contrast, although in low numbers, cestodes and trematodes were only present in sheep.

Our results show that sheep–goat mixed management can be considered as a risk factor for some of the most common parasite infections in small ruminants. Two situations should be considered. The first one concerns non-specific parasites, i.e., parasites shared between sheep and goats, such as lungworms and GIN; trematode and *Moniezia* infections can be discarded, since there have been no goats infected by these parasites. A possible explanation could be that compared to sheep, goats seem to develop a low immune response against parasitic nematodes. Due to their “browser” condition, goats have developed, comparatively to sheep, a low ability to mount an immune response to nematodes. This can explain why, under grazing conditions, goats do not develop resistance to GIN as efficiently as sheep [17]. Under this situation, it is easy to explain the effect of goats as risk factor over sheep under mixed management, with the goats functioning as a source of pasture contamination [13], but also contaminating feeders, drinking troughs and bedding.

This explanation is not valid for host-specific parasites, as is the case of *Eimeria* spp., which is rarely transmissible from one host species to another [18]; in fact, cross-infection experiments between sheep and goats have been unsuccessful [19]. So, although in some cases goats can excrete significantly higher oocyst output than sheep [20], it cannot be considered as a source of infection from oocysts to sheep. One explanation of the effect of mixed management, even if parasite species are different, could be that goats tend to occupy feeding areas with less parasitic burden; sheep are grass grazers and tend to graze high quality portions of the plant, whereas goats, as active foragers, tend to select highly digestible portions of grasses, but also ingest browse that is woody or stemmy and consume flowers, fruits and leaves, with less parasitic forms than sheep [21], and refuse water with fecal contamination or wet food with moldy smell [22]. Goats also have a strong hierarchical structure maintained by fighting [22]. It is easy to suppose that horned goats rule over sheep to define feeding areas. A pure flock could use a complete area to graze, but a mixed managed flock can suppose a reduced area for sheep, provoking even more contamination because of the concentration. Moreover, a situation of competing for food is certainly stressful, and stress plays a role in the susceptibility of lambs to coccidiosis, and also to acute coccidiosis in adult animals that have been subjected to stress [23].

## 5. Conclusions

This study mainly shows that mixed sheep–goat flocks are a risk factor that favors sheep parasitization; prevalence and, in a lesser extent, intensity of infection of different parasites in sheep can be favored by a mixed management of these animals with goats. On the other hand, it could be possible that mixed management could also affect different parasites in goats. Specific studies for each infection in sheep and goat flocks should consider the mixed management as a possible risk factor to be included in multivariate analyses.

## Figures and Tables

**Figure 1 animals-11-02077-f001:**
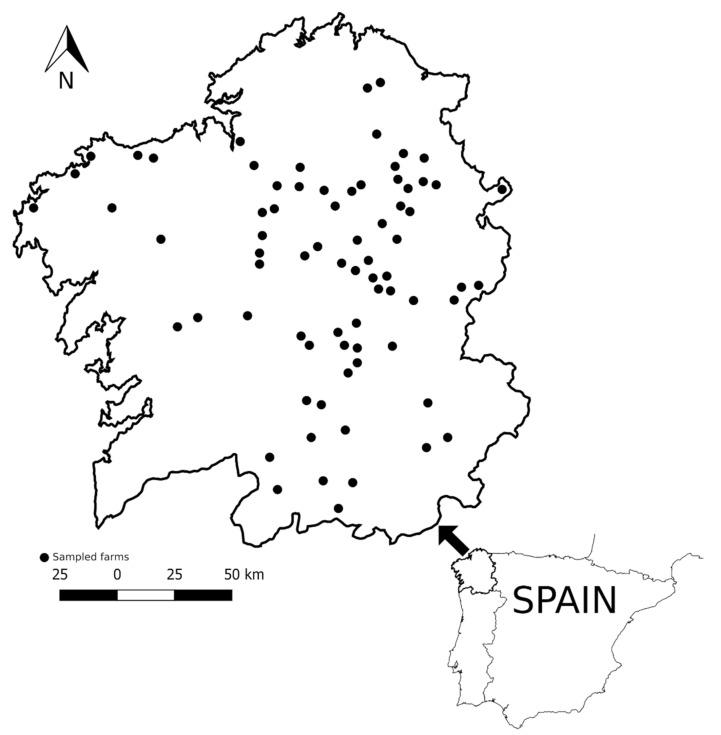
Geographical localization of sampled farms.

**Figure 2 animals-11-02077-f002:**
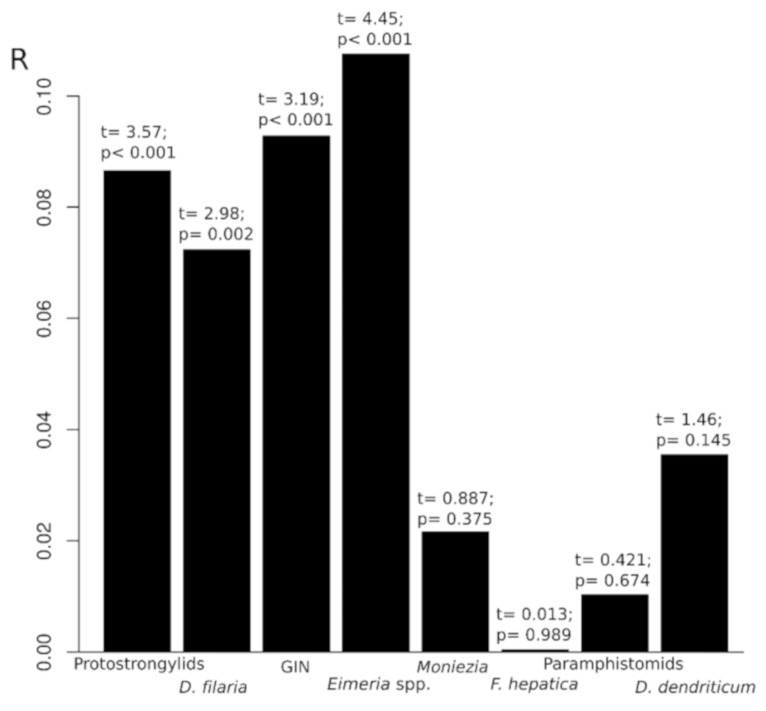
Partial least squares path modeling: correlation between mixed management and parasite infection in sheep.

**Table 1 animals-11-02077-t001:** Total prevalence and intensity of infection in sheep and goats in Galicia, NW Spain, for all parasites studied.

Parasitic Infection	Prevalence Positive/Total (%, 95% C.I. ^1^)		Intensity of Infection Mean (Parasitic Forms per g); sd	
	Goat	Sheep	Goat	Sheep
Protostrongylid nematodes	81/103 (78.6, 69.2–85.9)	242/2093 (11.6, 10.2–13.0)	283.2; 787.40	11.9; 30.85
*Dictyocaulus filaria*	11/103 (10.7, 5.7–18.7)	223/2093 (10.6, 9.4–12.1)	3.7; 6.36	8.5; 41.89
Gastrointestinal nematodes	75/103 (72.8, 63.0–80.9)	1233/1914 (64.4, 62.2–66.6)	634.7; 1507.07	465.7; 930.61
*Eimeria* spp.	86/103 (83.5, 74.6–89.8)	1415/1914 (73.9, 71.9–75.9)	2808.7; 6924.40	1393.3; 10,369.73
*Moniezia* spp.	0/103 (0, 0–0.04)	96/1914 (5.0, 4.1–6.1)	-	358.6; 700.37
*Fasciola hepatica*	0/103 (0, 0–0.04)	104/1697 (6.1, 5.1–7.4)	-	128.2; 209.23
*D. dendriticum*	0/103 (0, 0–0.04)	14/1697 (0.8, 0.5–1.4)	-	73.7; 64.97
Paramphistomidae	0/103 (0, 0–0.04)	12/1697 (0.7, 0.4–1.3)	-	89.2; 65.30

^1^ 95% Confidence interval.

**Table 2 animals-11-02077-t002:** Prevalence of parasitic infections in mixed and pure sheep flocks. Distribution analyses and OR depending on goat contact.

Parasitic Infection	Mixed Flocks Positive/Total (%, 95% C.I. ^1^)	Pure Sheep Flocks Positive/Total (%, 95% C.I.)	Chi-Squared Result	OR (95% C.I.)
Protostrongylid nematodes	83/468 (17.7, 14.4–21.6)	159/1625 (9.8, 8.4–11.4)	χ^2^ = 21.690; *p* < 0.001 ***^2^	1.99 (1.47–2.67)
*Dictyocaulus filaria*	71/468 (15.2, 12.1–18.8)	152/1625 (9.3, 8.0–10.9)	χ^2^ = 12.312; *p* < 0.001 ***	1.73 (1.26–2.37)
Gastrointestinal nematodes	300/399 (75.2, 70.6–79.3)	933/1515 (61.6, 59.1–64.0)	χ^2^ = 11.557; *p* < 0.001 ***	1.89 (1.46–2.45)
*Eimeria* spp.	322/399 (80.7, 76.4–84.4)	1093/1515 (72.1, 69.8–74.4)	χ^2^ = 11.997; *p* < 0.001 ***	1.61 (1.22–2.15)
*Moniezia* spp.	20/399 (5.0, 3.2–7.8)	76/1515 (5.0, 4.0–6.3)	χ^2^ = <0.001; *p* = 1	1 (0.57–1.68)
*Fasciola hepatica*	21/341 (6.2, 3.9–9.4)	83/1356 (6.1, 4.9–7.6)	χ^2^ = 0.004; *p* = 0.949	1.01 (0.58–1.67)
*D. dendriticum*	5/341 (1.5, 0.5–3.6)	9/1356 (0.7, 0.3–1.3)	χ^2^ = 1.276; *p* = 0.259	2.23 (0.58–7.45)
Paramphistomidae	3/341 (0.9, 0.2–2.8)	9/1356 (0.7, 0.3–1.3)	Fisher’s Test ^2^ *p* = 0.716	1.33 (0.23–5.36)

^1^ 95% Confidence interval; ***^2^ indicates significance level; probability of type I error less than 0.001. ^2^ Fisher’s exact test was applied to the Paramphistomidae distribution because, in this infection, the data matrix presented one cell out of four with less than 5 cases.

**Table 3 animals-11-02077-t003:** Intensity of infection in mixed and pure sheep flocks and results of Wilcoxon rank test depending on goat contact.

Parasitic Infection	Mixed Flocks Mean (Parasitic Forms per g); sd	Pure Sheep Flocks Mean (Parasitic Forms per g); sd	Wilcoxon Rank Test Result
Protostrongylid nematodes	15.9; 36.33	9.8; 27.42	W = 5546.5; *p* = 0.025 *^1^
*Dictyocaulus filaria*	6.6; 19.53	9.4; 49.00	W = 5844.5; *p* = 0.318
Gastrointestinal nematodes	521.9; 1121.16	447.7; 860.35	W = 137,910; *p* = 0.703
*Eimeria* spp.	1119.7; 4392.13	1474.0; 11555.95	W = 171,470; *p* = 0.484
*Moniezia* spp.	442.1; 931.06	334.1; 622.63	W = 839.0; *p* = 0.906
*Fasciola hepatica*	142.9; 160.36	124.6; 220.06	W = 720.5; *p* = 0.360
*Dicrocoelium dendriticum*	82.6; 46.59	68.7; 75.48	W = 17.5; *p* = 0.545
Paramphistomidae	168.3; 73.36	62.8; 37.24	W = 1.0; *p* = 0.026 *^1^

*^1^ indicates significance level; probability of type I error less than 0.05.

## Data Availability

Data are available from the corresponding author under reasonable request.

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
