# Peer review of "The Goat as a Risk Factor for Parasitic Infections in Ovine Flocks"

_animals, 2021, doi:10.3390/ani11072077_

Round 1
Reviewer 1 Report
1 The diagnosis of Parasitic infection has not been enough described clearly. How to distinguish different of egg of Parasite?
2 The study make a comparison of the parasite infections between mixed flocks and pure sheep flocks, and make a conclusion. However, I think before doing that,
risk factors related to animal and farmer status, farm and pasture management(vermifuge), and environmental factors (seasons, Temperatures, etc. ) should be examined for their association with the prevalence of parasite infections
3 The Similar conclusion to the one of the studies had been published. For example, Parasitic infection in goats was higher than that for sheep in a study in South Africa (Open Journal of Veterinary Medicine Vol. 2 No. 1 (2012)). The novelty of the study needs to be explored further.
4 The part of background should be strengthened, such as adding more relevant references.
Author Response
Thank you for the time in reading this manuscript. Your and comments were very much appreciated to improve the final version for publication.
1 The diagnosis of Parasitic infection has not been enough described clearly. How to distinguish different of egg of Parasite?
AU. We have used morphological techniques to diferenciate parasitic forms. We have included in line 106 Morphological diagnostic techniques to make it clear
2 The study make a comparison of the parasite infections between mixed flocks and pure sheep flocks, and make a conclusion. However, I think before doing that, risk factors related to animal and farmer status, farm and pasture management(vermifuge), and environmental factors (seasons, Temperatures, etc. ) should be examined for their association with the prevalence of parasite infections
AU. Of course; for many years we have been studying ruminant parasites in this region and we have observed this effect in small ruminants many times. Sometimes it has been included in scientific works (reference 2, over protostrongylids, and reference 3, over Visna Maedi virus), but in this case we have tried to analyze the most frequent parasites in sheep, all of them. Many of the factors have been studied by us as well as by many other authors and the influence is known. In this study we tried to highlight the mixed herd factor over the rest, since for each parasite the rest of the factors influence differently and it is very difficult to cover them all for nematodes, trematodes, cestodes and protozoa.
3 The Similar conclusion to the one of the studies had been published. For example, Parasitic infection in goats was higher than that for sheep in a study in South Africa (Open Journal of Veterinary Medicine Vol. 2 No. 1 (2012)). The novelty of the study needs to be explored further.
AU. We know this paper, but it does not indicate goat as a risk factor for sheep. It main conclusion is "The emerging profile of sheep and goats at high risk for GI helminth infections in Greece is that of animals belonging to farms where the educational level of the farmer is low and the farm is located in a mountainous area with moderate vegetation." That's why we did not included it in references. We did not have read other manuscript with goat as a risk factor for a group of parasites in sheep.
4 The part of background should be strengthened, such as adding more relevant references.
AU. Most of the references over small ruminants are refered to prevalences in different world areas. I think we cover the most important papers related to the factor studied in this manuscript, the mixed-management as a risk factor.
Reviewer 2 Report
The present investigation treats an interesting topi con risk factors related to parasitic infections among sheeo and goats.
However, some modifications are needed to improve the manuscript.
Abtract: In this section, the objective of the research should be mentioned as well as part of the methodology as well as the most relevant results and a small conclusion.
General comments: Based on which criteria they decided to analyze the parasites mentioned in the manuscript. It is not mentioned what previous handling both species had, such as deworming. Or did all the animals have the same sanitary management?
The authors do not mention the productive system in which the animals were found (intensive, semi-intensive or extensive).
Establish in which age range the study animals were found, since it has been shown that there may be a higher incidence of certain parasites in young animals.
Define whether sheep or goats act as intermediate or definitive host for the parasites examined in the research.
In results, both in table 1 and 2, the abbreviation (C.I.) is not defined.
In line 78 the abbreviation (GIN) is used for gastrointestinal nematodes, but in the tables it uses the full name and in figure 2. the abbreviation is used. I recommend that you consider the same term or abbreviation in the results.
Author Response
Thank you very much for your review and the comments you included to enhance this manuscript.
1. Abtract: In this section, the objective of the research should be mentioned as well as part of the methodology as well as the most relevant results and a small conclusion.
AU. We think the objective and conclusion are included ("...so the main objective of this study was to establish if mixed management with goats supposes a risk factor for parasitic infections in ovine flocks." and "...should consider sheep-goat mixed management as a possible risk factor to be included in multivariate analyses."). We are nor sure about methodology in abstract, as we are limited in characters.
2. General comments: Based on which criteria they decided to analyze the parasites mentioned in the manuscript. It is not mentioned what previous handling both species had, such as deworming. Or did all the animals have the same sanitary management? and The authors do not mention the productive system in which the animals were found (intensive, semi-intensive or extensive).
AU. We have included in lines 69-71 a paragraph that includes general management in this region and also treatment procedures.
3. Establish in which age range the study animals were found, since it has been shown that there may be a higher incidence of certain parasites in young animals.
AU. We try to sample a representative group of sheep and goats. To avoid the effect of very young animals over the intensity data, we sampled animals over 6 months (lines 78-79)
4. Define whether sheep or goats act as intermediate or definitive host for the parasites examined in the research.
AU. Sheep and goats are definive host of all of them but we have it as a normal knowledge. We do not know where we can include this in the manuscript.
5. In results, both in table 1 and 2, the abbreviation (C.I.) is not defined.
AU. Now 95% C.I. included and defined in tables 1 and 2
6. In line 78 the abbreviation (GIN) is used for gastrointestinal nematodes, but in the tables it uses the full name and in figure 2. the abbreviation is used. I recommend that you consider the same term or abbreviation in the results.
AU. We think that in tables full name must be used. In the case of Figure 2 GIN is used to avoid a very long category in x axis, that could be confusing
Reviewer 3 Report
The manuscript presents the results of the analysis if the sheep-goat mixed management is the risk factor for the main digestive and respiratory ovine parasitic infections.
In my opinion the manucript is well written. All statistical analysis are conducted properly which is imprtant in this kind of article. I have only two minor things for the authors which should be consider/improved:
- The conclusions are too general. Please, provide more information what these studies reveal and what is the message for readers.
- Please use the italics when you describe name of the parasite, for example in line 115: filaria, Eimeria spp. etc
Author Response
We would like to thank your comments, which have helped to make big
clarity in the manuscript.
- The conclusions are too general. Please, provide more information what these studies reveal and what is the message for readers.
AU. We have introduced a new phrase in the beginning of Conclusions (lines 209-210) to make it clearer: " This study mainly shows that mixed sheep-goat flocks are a risk factor that favors sheep parasitization; ...
2. Please use the italics when you describe name of the parasite, for example in line 115: filaria, Eimeria spp. etc
AU. We have changed all of them
Reviewer 4 Report
The paper describes a basic survey of parasites infecting sheep and goat and investigate the associate risk for infection between sheep and goat
In general the manuscript is well written, but there is shortcomings in the description and details of the M&M section
In the Introduction it is not immediately clear that co-grazing sheep and goats is an issue
L61 2 must be superscript.
L62 (and elsewhere) space between number and units
Fig 1 – More information needed on map. 1. Scale bar needed. 2. The location within Spain is not that obvious from the figure – darker lines to demarcate the area in the main map. Caption must be improved. Where is the farms (county/district), type of farms, etc…
L73 The authors refer to the procedure to detect lungworm larvae as “migration”, but do they not mean Baermann? The description of the technique seems incomplete. Is the funnel not filled with water for example? Suitable reference to technique required.
L75 details of make, supplier etc of centrifuge tube
L76 (and elsewhere) – use SI abbreviation of volumes, mass, etc….
L77 – similary need details of Favati chambers
L76 (and elsewhere) never start a sentence with a number
What was the minimum detection limit of the McMaster?
L80 – details on glass balls – size, make, manufacturer, etc….
L81/87 – details on 150 um sieve
L82 centrifugal force is written as 500 xg (the g is also in italics! Otherwise it means gram)
L83 – what is SG of salt
L84 – rephrase to Parasitic elements… and chamber must be plural (presumably both chambers have been examined)
L84 – presumably only strongyle type eggs were counted? Insert appropriate text
L88 – details on plastic cup
L 88 use SI abbreviation of volume (= 1 L)
L93 – it is not the usual way in which sedimentation test is examined. Is only the grid examined or whole chamber? Top or bottom? Only one aliquot examined? Either provide reference or more details on how level of infection is actually calculated.
L115 (and elsewhere) genus and specie names must be italicised.
L73 – was no attempt made to distinguish between protostongylid larvae? Why not?
Improve caption of Table 1 – more detail – of what, where….
In Tables presumably the % means 95% C.I.? Insert in column heading to make clear. Also CI abbreviation needs to be explained in footnote to table (apply to every table)
Table 1 – the Fasciola fec seems very high (especially for sheep with apparently no clinical signs). Brings me back to the M&M on how this was calculated
L131 – need p-value when claiming significance
L136-138 – rephrase
L138 – what is low?
Fig 2 – elsewhere the term Paramphistomidae is used. What does t – represent?
L155 ..coccidia
L205 – it is stated that the techniques were non-invasive. I strongly disagree – faecal samples were collected by rectum.
L 216 – replace recovered by accessed
A significant omission of this study is that the GIN infecting sheep and goats were not identified. Although they do share common parasites, some can be specie specific and from the results it is not clear.
Author Response
Thank you very much for your effort and the comments to enhance this research:
- In the Introduction it is not immediately clear that co-grazing sheep and goats is an issue
AU. We have included (Lines 69-71) a knew paragrapgh to make it clear
2. L61 2 must be superscript. - Changed
3. L62 (and elsewhere) space between number and units - corrected
4. Fig 1 – More information needed on map. 1. Scale bar needed. 2. The location within Spain is not that obvious from the figure – darker lines to demarcate the area in the main map. Caption must be improved. Where is the farms (county/district), type of farms, etc…
AU. A scale bar was added and Galician lines in Spain map were darker. Farm type has been described in a new paragraph in lines 69-71. We think that it is not necessary to include district, as infections are influenced more by plain/mountain/coast area, and this factor is not included in this study
5. L73 The authors refer to the procedure to detect lungworm larvae as “migration”, but do they not mean Baermann? The description of the technique seems incomplete. Is the funnel not filled with water for example? Suitable reference to technique required.
AU. Yes, it is a traditional Baermann-Wetzel technique. Baermann-Wetzel technique was included in line 83. We think that references are not necessary (Baermann 1917, Wetzel 1930), as this technique is a common knowlege for parasitologists
6. L75 details of make, supplier etc of centrifuge tube
AU. As we use thousands, we use old glass tubes from Vauntainer or Venojet. Sometimes one, others the other. We think that this is not important to reproduce the research study.
7. L76 (and elsewhere) – use SI abbreviation of volumes, mass, etc…. - changed
8. L77 – similary need details of Favati chambers
AU. Favati chambers are methacrylate hand-made chambers and there is not a specific one. As in tubes, this is not important to reproduce the research study, as in Favati chamber the specialist reads all volume
9. L76 (and elsewhere) never start a sentence with a number
AU. We cannot find in the manuscript any phrase with a number in the beginning
10. What was the minimum detection limit of the McMaster?
AU. It depends on the number of chambers readed in any study. In this case we read one chamber, included out of the grid, but only to point the animal as positive. Count was done only under the grid area. So, in flotation technique in this study, counts begin in 50 e.p.g
11. L80 – details on glass balls – size, make, manufacturer, etc….
AU. As in tubes and Favati chambers, we buy them in bags, with different origins, but they are only glass balls; no influence in final results; they only break faeces, and this if confirmed in 150 um sieves. Balls diameter is 5-6 mm
12. L81/87 – details on 150 um sieve
AU. As in tubes and balls, we use different types; all of them guarantee pore width (metallic structure, nylon lines in 150nm and below). In any case, we can guarantee functionality, as we use them and prove the effectivity in the technique with any new sieve in its first use; in any case, this cannot influence over reproducing the research study
13. L82 centrifugal force is written as 500 xg (the g is also in italics! Otherwise it means gram) - all cases changed to italic g
14. L83 – what is SG of salt - included in text
15. L84 – rephrase to Parasitic elements… and chamber must be plural (presumably both chambers have been examined).
AU. We think that McMaster chamber is considered as ONE, with two compartments
16. L84 – presumably only strongyle type eggs were counted? Insert appropriate text
AU. We use parasitic forms because we count in flotation not only strongyle eggs, but also cestode eggs and protozoal cysts (Eimeria). We try to include all of them in one sentence
17. L88 – details on plastic cup
AU. Plastic cup in this case is refered to a conical plastic cup thar favors egg decantation. To make it clearer, we have included conical cup in line 100
18. L 88 use SI abbreviation of volume (= 1 L) - changed
19. L93 – it is not the usual way in which sedimentation test is examined. Is only the grid examined or whole chamber? Top or bottom? Only one aliquot examined? Either provide reference or more details on how level of infection is actually calculated.
AU. We read one chamber, in the bottom, including out of the grid. to make it clearer we include in line 105 in the bottom of the McMaster chamber.
20. L115 (and elsewhere) genus and specie names must be italicised. - Corrected
21. L73 – was no attempt made to distinguish between protostongylid larvae? Why not?
AU. Yes, we distinguish protostrongylid larvae; we have years of experience in morphological diagnoses, but because of treatments, we find almost only M. capillaris (Lopez et al, 2011 and so on) in last years and we think that small percentages of other species are not important for this study
22. Improve caption of Table 1 – more detail – of what, where….
AU. Table 1 caption changed to Total prevalence and intensity of infection in sheep and goats in Galicia, NW Spain, for all parasites studied
23. In Tables presumably the % means 95% C.I.? Insert in column heading to make clear. Also CI abbreviation needs to be explained in footnote to table (apply to every table)
AU. Yes, of course. Changes to 95% C.I. Abbreviation explained in Table 1 and 2
24. Table 1 – the Fasciola fec seems very high (especially for sheep with apparently no clinical signs). Brings me back to the M&M on how this was calculated
AU. We do not thing 128 e.p.g. is too much. In Galicia, with a humid climate, Fasciola is very common in cattle (almost 1 million in 29.000 km2), so soils are contaminated. On the other hand, prevalence was only 6.1% and we only read one chamber. This means that low infections could not be detected, increasing mean elimination
25. L131 – need p-value when claiming significance - P value included
26. L136-138 – rephrase
AU. English expert recommended two slight changes in lines now 150-153
27. L138 – what is low?
AU. As indicated by figure 2, R value is around 0.1, indicating that there are another variables that affect infection, as we say in this paragraph.
28. Fig 2 – elsewhere the term Paramphistomidae is used. What does t – represent?
AU. With Paramphistomids we try to say that there should be more than one paramphistomid in Galicia. We detect continuosly two different type (in length) of paramphistomid eggs (we are now trying to detect if there are more than one)
t represent the statistical estimator of the algorith of plspf() analysis.
29. L155 ..coccidia - corrected
30. L205 – it is stated that the techniques were non-invasive. I strongly disagree – faecal samples were collected by rectum.
AU. For our regulation, faecal smpling is not invasive and it is easy to obtain permission to study under this sampling conditions. Blood sampling, for instance, is considered invasive and it is much more difficult to obtain permission
31. L 216 – replace recovered by accessed - replaced
32. A significant omission of this study is that the GIN infecting sheep and goats were not identified. Although they do share common parasites, some can be specie specific and from the results it is not clear.
AU. Yes. In fact we identified some of the samples in pools, but not all of them, so genera identification cannot be used as individual variable. No time or workforce to do it in all cases
Round 2
Reviewer 4 Report
The authors seem to want to skip over crucial important details in the M&M that I regard as important in a good scientific paper
L70-71. This is rather vague. It is important to indicate approximately when last animals were treated. This can have a significant effect on the results of the study. If not known – state that in the manuscript.
L83. I disagree. The Baermann-Wetzel technique maybe common but will not necessarily be familiar to students and or non-parasitologists. Also, there are many variations of the technique, so it is important that the appropriate reference be supplied (and to the technique as performed in this study). Similar to what have been done with the McMaster technique as this is also a well known procedure
L89, 96 Sentences start with numeral.
L87 correct unit for centrifugal force: xg (g in italics)
L87. Again, I disagree. I have never come across the term “Favati chambers” perhaps there is another more common equivalent, so the authors either need to provide suitable reference or describe the chambers in more details. Is it similar to a watch glass (https://www.preproom.org/equipment/eq.aspx?eqID=5077) or staining dish?
L93. Specific gravity is indicated by SG 1.19 (with no units)
L89. The description of the detection limit of the McMaster is not clear. Need further clarification in M&M
L90. Insert size of glass balls
L91, 98. Insert the details of one of the sieves used in this study.
L89, 104. “We think that McMaster chamber is considered as ONE, with two compartments” This terminology is confusing. It is commonly referred to as a McMaster slide with chambers/compartments. McMaster slides are commonly available with either two or three chambers. Also see the reference cited (4) how it describes the McMaster slide. So the M&M can be made clearer
L89. For the GIN it must be indicated that it is strongyle type eggs that were counted (which would exclude eggs such as Strongyloides, Trichuris, Nematodirus, etc)
L 104. For the sedimentation it is still not clear how the epg was calculated.
“Yes, we distinguish protostrongylid larvae; we have years of experience in morphological diagnoses”, I think this result is important to mention in the manuscript - we find almost only M. capillaris ……
Author Response
Thank you for you effort to enhance this manuscript,
L70-71. This is rather vague. It is important to indicate approximately when last animals were treated. This can have a significant effect on the results of the study. If not known – state that in the manuscript.
AU. Now Lines 79-81. We have included this sentence "Sampled animals had not received an anthelmintic treatment since the previous campaign, that is, animals were sampled at least two months after the last treatment."
L83. I disagree. The Baermann-Wetzel technique maybe common but will not necessarily be familiar to students and or non-parasitologists. Also, there are many variations of the technique, so it is important that the appropriate reference be supplied (and to the technique as performed in this study). Similar to what have been done with the McMaster technique as this is also a well known procedure
AU. [3] Baermann G. Eine einfache Methode zur Auffindung von Ancylostomum-(Nematoden)-Larven in Erdproben. Geneesk. Tijaschr. Nederl.-Indië, 1917, 57, 131-137 and [4] Wetzel R. Zur Diagnose der Lungenwurminvasionen bei Rind und Schaf. Dtsch. Tierärztl. Wschr., 1930, 38, 49-50. included in text and references
L89, 96 Sentences start with numeral.
AU. Corrected (Yes, there were two sentences with number!)
L87 correct unit for centrifugal force: xg (g in italics)
AU. All of the centrifugal g were corrected yesterday! I am sure of it! I have checked it
L87. Again, I disagree. I have never come across the term “Favati chambers” perhaps there is another more common equivalent, so the authors either need to provide suitable reference or describe the chambers in more details. Is it similar to a watch glass (https://www.preproom.org/equipment/eq.aspx?eqID=5077) or staining dish?
AU. This sentence has been included in lines 91-93 "These methacrylate counting chamber present a cell of 24 x 24 x 3 mm, enabling the microscopic observation of the volume resulting from centrifugation; the cell contain a permanent grid that allows the user to count more accurately."
L93. Specific gravity is indicated by SG 1.19 (with no units)
AU. 1.19 g/ml at 20 ºC changed to SG 1.19 at 20 ºC
L89. The description of the detection limit of the McMaster is not clear. Need further clarification in M&M
AU. This sentence "... the number of eggs per g of feces is obtained by multiplying the total number of eggs in the two squares by 50 (3 g of feces yielded 45 ml of suspension and 0.3 ml examined [6]." was added to lines 101-3. It is similar to that described in [6] MAFF manual
L90. Insert size of glass balls
AU. (6 mm diameter) included after glass balls
L91, 98. Insert the details of one of the sieves used in this study.
AU. Company of production included in line 97 "(CISA Sieving Techlogies, Barcelona)"
L89, 104. “We think that McMaster chamber is considered as ONE, with two compartments” This terminology is confusing. It is commonly referred to as a McMaster slide with chambers/compartments. McMaster slides are commonly available with either two or three chambers. Also see the reference cited (4) how it describes the McMaster slide. So the M&M can be made clearer
AU. We think that the sentence included in lines 101-3 makes it clear now (three numbers back).
L89. For the GIN it must be indicated that it is strongyle type eggs that were counted (which would exclude eggs such as Strongyloides, Trichuris, Nematodirus, etc)
AU. (GIN - strongyle type eggs) added in line 92
L 104. For the sedimentation it is still not clear how the epg was calculated.
AU. Eggs are calculated in a similar way that flotation (but in the bottom of the chamber), so we think that it is not neccessary to repeat the calculation
“Yes, we distinguish protostrongylid larvae; we have years of experience in morphological diagnoses”, I think this result is important to mention in the manuscript - we find almost only M. capillaris ……
AU. We include "(mainly Muellerius capillaris, but for 4 sheep with Neostrongylus linearis)" just after protostrongylids Line 86